# Genetic diversity and population structure of indigenous chicken in Rwanda using microsatellite markers

**Richard Habimana**[1,2]*, **Tobias Otieno Okeno**[2], **Kiplangat Ngeno**[2], **Sylvere Mboumba**[3], **Pauline Assami**[4], **Anique Ahou Gbotto**[5], **Christian Tiambo Keambou**[4,6], **Kizito Nishimwe**[1], **Janvier Mahoro**[1], **Nasser Yao**[4]

**1** Animal Science and Veterinary Medicine, University of Rwanda, College of Agriculture, Nyagatare, Rwanda, **2** Department of Animal Science, Animal Breeding and Genomics Group, Egerton University, Nakuru, Kenya, **3** Faculty of Agronomy and Biotechnologies, Masuku University of Science and Techniques, Franceville, Gabon, **4** Biosciences eastern and central Africa, International Livestock Research Institute Hub, Nairobi, Kenya, **5** Laboratoire de genomique Fonctionnelle et Amélioration Génétique Université Nangui Abrogoua, Abidjan, Cote d'Ivoire, **6** Faculty of Agriculture and Veterinary medicine, University of Buea, Buea, Cameroon

* hrichard86@yahoo.fr

**Data Availability Statement:** All relevant data are within the manuscript and its Supporting Information files.

## Abstract

Rwanda has about 4.5 million of indigenous chicken (IC) that are very low in productivity. To initiate any genetic improvement programme, IC needs to be accurately characterized. The key purpose of this study was to ascertain the genetic diversity of IC in Rwanda using microsatellite markers. Blood samples of IC sampled from 5 agro-ecological zones were collected from which DNA was extracted, amplified by PCR and genotyped using 28 microsatellite markers. A total of 325 (313 indigenous and 12 exotic) chickens were genotyped and revealed a total number of 305 alleles varying between 2 and 22 with a mean of 10.89 per locus. One hundred eighty-six (186) distinct alleles and 60 private alleles were also observed. The frequency of private alleles was highest in samples from the Eastern region, whereas those from the North West had the lowest. The influx of genes was lower in the Eastern agro-ecological zone than the North West. The mean observed heterozygosity was 0.6155, whereas the average expected heterozygosity was 0.688. The overall inbreeding coefficient among the population was 0.040. Divergence from the Hardy-Weinberg equilibrium was significant (p<0.05) in 90% of loci in all the populations. The analysis of molecular variance revealed that about 92% of the total variation originated from variation within populations. Additionally, the study demonstrated that IC in Rwanda could be clustered into four gene groups. In conclusion, there was considerable genetic diversity in IC in Rwanda, which represents a crucial genetic resource that can be conserved or optimized through genetic improvement.

**Funding:** This study was part of PhD research of the first author and he is thankful to the financial and technical support from BecA-ILRI Hub through Africa Biosciences Challenge Fund (ABCF) programmes. The ABCF Programmes is funded by the Australian Department for Foreign Affairs and Trade (DFAT) through the BecA-CSIRO partnership; the Syngenta Foundation for Sustainable Agriculture (SFSA); the Bill & Melinda Gates Foundation (BMGF); the UK Department for International Development (DFID) and the Swedish International Development Cooperation Agency (SIDA). This material is also based upon work supported by the United States Agency for International Development, as part of the Feed the Future initiative, under the CGIAR Fund, award number BFS-G-11-00002, and the predecessor fund the Food Security and Crisis Mitigation II grant, award number EEM-G-00-04-00013. The funders had no role in study design, data collection and analysis, decision to publish, or preparation of the manuscript.

**Competing interests:** The authors have declared that no competing interests exist.

## Introduction

Poultry keeping is an agricultural enterprise with a high potential in Rwanda. More than 40% of households keep poultry with indigenous chickens (IC) being the most preferred, accounting for approximately 80% of the reared chicken species [1]. Raising IC is preferred to exotic breeds because of their small cost of production, scavenging capacity and adaptability to harsh environmental conditions. IC production serves a critical role as a source of revenue for resource-limited countryside families [2]. However, the productivity of IC in Rwanda is low. Each mature hen weighs between 0.8 to 1.8 kg and produces an average of 40 to 100 eggs per year. This output is insufficient to meet the needs of the population [3] and mitigate poverty among the smallholder farmers in rural areas. To improve the genetic potential of IC in Rwanda, different crossbreeding programmes between IC and exotic chicken have been initiated. However, these programmes have not been sustainable due to decreased broodiness in the hybridized birds, unpredictable stock, and the high cost of buying and sustaining exotic cocks for breeding purposes. Additionally, recent global efforts to preserve native genetic resources pose a threat to such programmes [4]. There is, therefore, a need for the development of an alternative strategy to genetic improvement and conservation of IC.

Genetic improvement through within-breed selection of IC in Rwanda could be a promising alternative strategy. Nonetheless, genetic enhancements need a resolute breeding objective, sustainable breeding plans, and an in-depth comprehension of the genetic diversity of prevailing genotypes and ecotypes [5]. Therefore, elucidating the genetic characteristics of the prevailing IC stock will not only favor genetic enhancement but will also expedite their preservation [4].

In various parts of the world, the genetic diversity of IC has been assessed using molecular markers including microsatellites [6–19]. Microsatellites are short, tandemly repeated simple sequences with one to six base pairs in length [20]. Thirty (30) microsatellite markers have been suggested by the Food and Agriculture Organization to be used in the evaluation of genetic diversity in chicken [20–21]. These microsatellite markers are appropriate for a wide range of applications and have remained the most commonly used markers in studies of genetic diversity and population structure since the early 1990s [20,22,23] due to their high degree of polymorphism, random distribution across the genome, codominance, and neutrality with respect to selection [24]. Additionally, they are relatively cheaper to genotype and offer more population genetic information per marker than single nucleotide polymorphisms (SNPs) known as biallelic markers [25]. Finally, microsatellites can successfully amplify low DNA concentration or low-quality DNA samples [26].

There is, however, a scarcity of data on the genetic diversity and population structure of IC in Rwanda. The availability of such knowledge could drive the understanding of the origin and genetic variability in the population to guide selection decisions. As a result, it would be possible to develop apposite mating plans to uphold genetic variation and minimize inbreeding in the population, which would promote response to selection. This study evaluated the degree of genetic diversity and phylogenetic relationships between populations of IC in Rwanda using simple sequence repeats (SSR) markers.

## Materials and methods

### Ethical statement

After a thorough review and approval of sampling procedures and experimental manipulations, ethical permission (Ref: 031/19/DRI September 2, 2019) for the collection of chicken blood samples was obtained from the Research Screening and Ethical Clearance Committee of

the College of Agriculture, Animal Sciences and Veterinary Medicine, University of Rwanda. Private grounds were never entered without the consent of chicken owners. The owners of the chicken signed an informed consent form to allow collection of blood sample from their chicken to be used for the experiment. A memorandum of understanding between University of Rwanda, Rwanda Agriculture and Animal Resources Development Board and Ministry of Agriculture had been made to oversee research and consent research activities including procedures to be undertaken in the whole country. Therefore, no specific permissions were needed for each location visited. Every zone was visited in a company of Rwanda Agriculture and Animal Resources Development Board employee who ensured that national and international guidelines were followed. In addition, the chickens were treated humanely, and none of them was sacrificed for this study.

## Collection of samples and DNA extraction

In total, 313 distinct IC, previously characterized morphologically (S1 Table) [27], were sampled from five agro-ecological zones [51, 52, 53, 55, and 102 were sampled from Central South (CS), North West (NW), Central North (CN), South West (SW), and East (E), respectively] (S1 Fig and Fig 1). Indigenous Chicken populations were reckoned according to agro-ecological zones [28]. Households having IC were randomly selected considering a minimum distance of 500 meters between them to ensure sampling of unrelated birds [29]. Twelve (12) exotic commercial chicken breeds (2 kuroilers, 5 Isa brown layers and 5 cobb broilers) were included as references. These exotic breeds have been developed from several parent breeds which are not usually divulged by breeder companies and, therefore, are marketed as commercial hybrids under trade names. They were genetically selected for performance traits associated with egg (layers), meat (broilers) or both egg and meat (kuroilers) production (S1 Table).

A single blood drop was drawn from veins in the wing of each bird and placed on Whatman FTA™ filter cards, left to dry in a cool place for approximately one hour, and held in reserve in discrete envelopes at room temperature awaiting further processing. The isolation of genomic DNA was done using Smith and Burgoyne's boiling method [30]. The quality of genomic DNA was ascertained through gel electrophoresis using 1% agarose. A NanoDrop Spectrophotometer (Thermo Scientific ™ Nanodrop 2000) was used to quantify the total DNA, which was adjusted to 10ng/μl before use in the subsequent steps of polymerase chain reaction (PCR) and genotyping.

## PCR amplification and DNA polymorphism

Twenty-eight fluorescently-labelled polymorphic SSR markers were chosen based on the extent of polymorphism shown by a high polymorphism information content and the genome coverage consistent across previous studies [31]. The PCR reactions had a total volume of 10μl consisting of 30ng target DNA, 5μl of One Taq 2MM and 0.2μl of each forward and reverse primer. The amplifications were done in a thermocycler (Applied Biosystems 9700 Thermal Cycler Gene Amp®) and entailed the first denaturation at 94˚C for 3 minutes, 30 cycles of denaturation at 94ºC for 30 seconds, the primer annealing at temperatures ranging between 58˚C and 64˚C based on the primer components (Table 1) for 1 minute, and extension at 72˚C for 2 minutes. The last extension step was done at 72˚C for 10 minutes. The PCR products of different fluorescent tags were combined according to the exhibited colour and intensity of bands to create uniform signal strength. Hi-Di formimide was used to denature the combined amplicons at 95˚C for 3 minutes, this step was followed by capillary electrophoresis separation in an ABI3730 DNA genetic analyzer by using GeneScan- 500 Internal LIZ and 1200 Internal

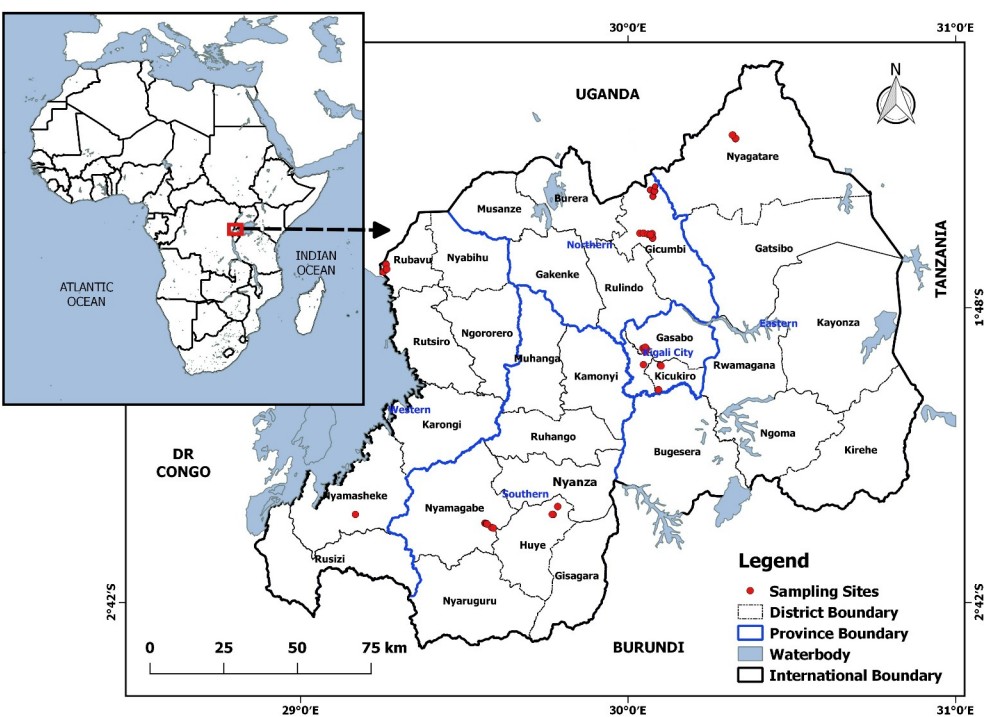

**Fig 1. Map of sampling sites of chicken blood used in this study.**

LIZ Size Standards. The resultant fragment analysis data and sizes of alleles were counted using GENEMAPPER software v. 4.1 (Applied Biosystems).

## Statistical analysis

**Genetic diversity and relationship.** The polymorphism information content (PIC) was estimated using Powermarker v.3.25 [6]. GenAlEx v.6.5 was used to estimate the allele frequencies, total alleles, expected heterozygosity (He), observed heterozygosity (Ho), and Wright's F-statistics as well as other parameters such as inbreeding coefficient over all populations (Fis), among populations (Fit) and within populations (Fst) for 28 microsatellite markers [7]. Jackknifing across populations using FSTAT v.2.9.4 produced standard deviation values that were used to obtain tests of significance per microsatellite locus by creating confidence intervals at 95% and 99% [8].

GENETIX v.4.05.2 was used to estimate genetic variation per breed (He, Ho) and the average number of alleles [9]. Gene flow [10] was calculated using Powermarker v.3.25 [6]. Pairwise Fst values, which are indications of the fraction of genetic variation attributed to population sub-structuring, were calculated for various population pairs using GenAlEx v.6.5 [7]. Analysis of molecular variance (AMOVA) was computed using GenAlEx v.6.5 for within and among pre-grouped populations [7]. Powermarker v 3.25 was used to assess genotype frequencies for nonconformity with Hardy-Weinberg equilibrium (HWE) in addition to linkage disequilibrium by performing Pearson's chi-squared test $(\chi)^2$ [6]. GenAlEx v.6.5 [7] was used to approximate Nei's standard genetic distances [11] among population pairs. The Neighbour-Joining (NJ) programme was used to develop an unrooted NJ cladogram using the Darwin software v.6.0 according to pairwise kinship distance matrix between populations [12]. A consensus tree assessed by 1,000 bootstraps all through the group of loci was created.

**Table 1. Sequences and physical information of 28 SSR markers used for PCR amplification.**

| Nam | Allele size (base-pairs) | Forward Primer 5'- 3' | Reverse primer 3'-5' | Annealing temperature (Tm: °C) |
|---|---|---|---|---|
| ADL0268 | 102–116 | CTCCACCCCTCTCAGAACTA | CAACTTCCCATCTACCTACT | 60 |
| MCW0206 | 221–249 | ACATCTAGAATTGACTGTTCAC | CTTGACAGTGATGCATTAAATG | 60 |
| LEI0166 | 354–370 | CTCCTGCCCTTAGCTACGCA | TATCCCCTGGCTGGGAGTTT | 60 |
| MCW0295 | 88–106 | ATCACTACAGAACACCCTCTC | TATGTATGCACGCAGATATCC | 60 |
| MCW0081 | 112–135 | GTTGCTGAGAGCCTGGTGCAG | CCTGTATGTGGAATTACTTCTC | 60 |
| MCW0014 | 164–182 | TATTGGCTCTAGGAACTGTC | GAAATGAAGGTAAGACTAGC | 58 |
| MCW0183 | 296–326 | ATCCCAGTGTCGAGTATCCGA | TGAGATTTACTGGAGCCTGCC | 58 |
| ADL0278 | 114–126 | CCAGCAGTCTACCTTCCTAT | TGTCATCCAAGAACAGTGTG | 60 |
| MCW0067 | 176–186 | GCACTACTGTGTGCTGCAGTTT | GAGATGTAGTTGCCACATTCCGAC | 60 |
| MCW0104 | 190–234 | TAGCACAACTCAAGCTGTGAG | AGACTTGCACAGCTGTGTACC | 60 |
| MCW0123 | 76–100 | CCACTAGAAAAGAACATCCTC | GGCTGATGTAAGAAGGGATGA | 60 |
| MCW0330 | 256–300 | TGGACCTCATCAGTCTGACAG | AATGTTCTCATAGAGTTCCTGC | 60 |
| MCW0165 | 114–118 | CAGACATGCATGCCCAGATGA | GATCCAGTCCTGCAGGCTGC | 60 |
| MCW0069 | 158–176 | GCACTCGAGAAAACTTCCTGCG | ATTGCTTCAGCAAGCATGGGAGGA | 60 |
| MCW0248 | 205–225 | GTTGTTCAAAAGAAGATGCATG | TTGCATTAACTGGGCACTTTC | 60 |
| MCW0111 | 96–120 | GCTCCATGTGAAGTGGTTTA | ATGTCCACTTGTCAATGATG | 60 |
| MCW0020 | 179–185 | TCTTCTTTGACATGAATTGGCA | GCAAGGAAGATTTTGTACAAAATC | 60 |
| MCW0034 | 212–246 | TGCACGCACTTACATACTTAGAGA | TGTCCTTCCAATTACATTCATGGG | 60 |
| LEI0234 | 216–364 | ATGCATCAGATTGGTATTCAA | CGTGGCTGTGAACAAATATG | 60 |
| MCW0103 | 266–270 | AACTGCGTTGAGAGTGAATGC | TTTCCTAACTGGATGCTTCTG | 64 |
| MCW0222 | 220–226 | GCAGTTACATTGAAATGATTCC | TTCTCAAAACACCTAGAAGAC | 60 |
| MCW0016 | 162–206 | ATGGCGCAGAAGGCAAAGCGATAT | TGGCTTCTGAAGCAGTTGCTATGG | 60 |
| MCW0037 | 154–160 | ACCGGTGCCATCAATTACCTATTA | GAAAGCTCACATGACACTGCGAAA | 64 |
| MCW0098 | 261–265 | GGCTGCTTTGTGCTCTTCTCG | CGATGGTCGTAATTCTCACGT | 60 |
| LEI0094 | 247–287 | GATCTCACCAGTATGAGCTGC | TCTCACACTGTAACACAGTGC | 60 |
| MCW0284 | 235–243 | GCCTTAGGAAAAACTCCTAAGG | CAGAGCTGGATTGGTGTCAAG | 60 |
| MCW0078 | 135–147 | CCACACGGAGAGGAGAAGGTCT | TAGCATATGAGTGTACTGAGCTTC | 60 |
| LEI0192 | 244–370 | TGCCAGAGCTTCAGTCTGT | GTCATTACTGTTATGTTTATTGC | 60 |
| ADL0112 | 120–134 | GGCTTAAGCTGACCCATTAT | ATCTCAAATGTAATGCGTGC | 58 |
| MCW0216 | 139–149 | GGGTTTTACAGGATGGGACG | AGTTTCACTCCCAGGGCTCG | 60 |

Source: FAO [32]

**Population structure.** The possible sum of clusters was approximated using the Evanno method [13] as reported by Dent Earl and Bridgett [14]. A set of rules applied in STRUCTURE v.2.3.4 was used to group entities based on multi-locus genotypes [15]. The evaluation entailed an admixture model alongside interrelated allele frequencies. During the STRUCTURE analysis, 5 replications of K (presumed sum of subpopulations), extending from 1 to 20 were used together with 100,000 reiterations of Markov Chain Monte Carlo (MCMC) and 50,000 burn-in period in the admixture model. Each estimation of K was redone 5 times to ensure the reproducibility of the outcomes. CLUMPAK (CLUMPAK server), which is a tool used to single out clustering types and bundle population structure deductions across K was used. The Factorial Correspondence Analysis (FCA), which is a multivariate model of analysis, was conducted to observe the associations between entities from unlike zones and to evaluate probable admixtures between the populations. The main variables were the frequencies of alleles at all loci in the populations. The FCA was computed using GENETIX v.4.05.2 [9].

## Results

### Genetic diversity

**Marker polymorphism across the studied IC populations.** The parameters of the variability of the investigated loci are shown in Table 2. Overall, 305 alleles were observed at the 28 microsatellite loci with an average of 10.89 alleles per microsatellite marker. The total sum of alleles ranged from 2 (MCW0037) to 22 (LEI0192). The effective number of alleles (NE) ranged between 1.6504 (MCW0078) and 8.901 (LEI0234), with an overall mean of 3.8194. The PIC ranged from 0.3488 (MCW0103) to 0.8775 (LEI0234). Out of the total number of alleles, 20% were private alleles (60), whereas ADL0112 revealed the maximum sum of private alleles (6). The within-population insufficiency in heterozygosity (as determined by $F_{IS}$ factor), extended between −1.00 (MCW0037) and 0.338 (LEI0234) with a mean of 0.041 for all loci. The inbreeding coefficient among populations ($F_{IT}$) values ranged from -1.00 (MCW0037) to 0.354 (LEI0234), with a mean of 0.089. Global population differentiation (evaluated by $F_{ST}$) was estimated at 0.054. The contribution of 28 microsatellites for population segregation (determined by $F_{ST}$ statistics) varied from 0.000 (MCW0037) to 0.158 (ADL0268). The overall F-statistics differed significantly (p<0.05) from zero. This differentiation had a significant contribution from all loci. The values for Ho ranged from 0.3015 (MCW0165) to 1 (MCW0037), with an overall mean of 0.6155, while the values of He ranged from 0.394 (MCW0078) to 0.8877 (LEI0234), with a general mean of 0.688. The average number of migrants per generation (Nm) in the whole population and across all the loci was found to be 6.06. Only 10% of the loci in all IC populations, did not differ considerably (p >0.05) from the HWE.

**Genetic diversity indices for IC populations from each agro-ecological zone.** Genetic diversity indices for IC from each zone is summarized in Table 3. All the loci were polymorphic. The observed and expected frequencies of heterozygote were not statistically different (p>0.05), hence, the inbreeding coefficient (F) estimates observed were not substantially different from zero. The mean sum of alleles varied from 5.143 to 8.25. The highest count of alleles (8.2) was found in the Eastern IC population. The highest count of private alleles (21) was observed in the Eastern population, while the NW population did not harbor any private allele. The effective sum of alleles ranged from 3.311 to 3.62. The Shannon Index (I), which is an expression of population diversity in a particular habitat, was high in the SW (1.458) and low in exotic chicken (1.305). Furthermore, the lowest observed heterozygosity was in the CS (0.598) while the highest was recorded in exotic chicken population (0.667). The expected heterozygosity in the populations ranged from 0.644 (CN) to 0.680 (SW).

The p-values of HWE are summarized in Table 4 and confirm that Ho and He did not differ significantly (P>0.05). Thus, taking all the loci into account none of the IC populations diverged from the HWE law.

Analysis of molecular variance revealed that ninety-two percent (92%) of the total variation originated from variation within populations (Table 5).

**Genetic relationship.** The matrix of pairwise genetic distances between populations (Table 6 and Fig 2) showed low genetic distance (0.029) between NW and CN populations. A similar trend was observed in SW and CS (0.048). On the other hand, by considering only the IC populations, the highest genetic distance was observed between E and SW populations (0.125). The genetic distance between the IC population in Rwanda and exotic chicken was relatively high (0.231).

The phylogenetic relationship by the Neighbour-Joining tree showed four (4) IC genetic clusters, namely I, II, III and IV (Fig 3). The eastern population stands alone unlike the other populations: IC populations from the NW clustered together with those from the CN. Few

**Table 2. Marker polymorphism and diversity parameters across studied IC populations in Rwanda.**

| Loci | MAF | NG | NA | NE | NPA | He | Ho | PIC | I | Fis | Fit | Fst | Nm | HWE pV |
|---|---|---|---|---|---|---|---|---|---|---|---|---|---|---|
| ADL0112 | 0.499 | 27 | 16 | 2.720 | 6 | 0.632 | 0.594 | 0.572 | 1.318 | 0.097 | 0.128 | 0.034 | 7.006 | 0.000 |
| ADL0268 | 0.245 | 39 | 14 | 6.241 | 3 | 0.840 | 0.582 | 0.820 | 2.022 | 0.176 | 0.306 | 0.158 | 1.332 | 0.000 |
| ADL0278 | 0.300 | 39 | 12 | 5.349 | 4 | 0.813 | 0.548 | 0.789 | 1.885 | 0.252 | 0.283 | 0.041 | 5.869 | 0.000 |
| LEI0094 | 0.392 | 45 | 17 | 4.360 | 3 | 0.771 | 0.714 | 0.744 | 1.867 | 0.017 | 0.034 | 0.017 | 14.344 | 0.000 |
| LEI0192 | 0.317 | 66 | 22 | 5.699 | 4 | 0.825 | 0.775 | 0.806 | 2.149 | -0.005 | 0.036 | 0.041 | 5.829 | 0.000 |
| LEI0234 | 0.177 | 77 | 17 | 8.902 | 2 | 0.888 | 0.569 | 0.878 | 2.393 | 0.338 | 0.354 | 0.024 | 10.202 | 0.000 |
| MCW0014 | 0.512 | 29 | 10 | 3.107 | 1 | 0.678 | 0.486 | 0.645 | 1.493 | 0.142 | 0.263 | 0.142 | 1.517 | 0.000 |
| MCW0016 | 0.317 | 39 | 15 | 4.699 | 4 | 0.787 | 0.772 | 0.759 | 1.841 | 0.002 | 0.023 | 0.021 | 11.392 | 0.000 |
| MCW0020 | 0.305 | 29 | 8 | 4.661 | 0 | 0.785 | 0.720 | 0.753 | 1.676 | 0.050 | 0.095 | 0.047 | 5.027 | 0.000 |
| MCW0034 | 0.351 | 46 | 14 | 5.211 | 5 | 0.808 | 0.775 | 0.788 | 1.927 | -0.003 | 0.032 | 0.035 | 6.965 | 0.191 |
| MCW0037 | 0.500 | 1 | 2 | 2.000 | 0 | 0.500 | 1.000 | 0.375 | 0.693 | -1.000 | -1.000 | 0.000 | | 0.000 |
| MCW0067 | 0.395 | 31 | 11 | 3.573 | 1 | 0.720 | 0.680 | 0.679 | 1.622 | 0.038 | 0.137 | 0.103 | 2.181 | 0.000 |
| MCW0069 | 0.339 | 26 | 10 | 3.671 | 0 | 0.728 | 0.739 | 0.680 | 1.503 | -0.011 | 0.028 | 0.038 | 6.309 | 0.104 |
| MCW0078 | 0.766 | 11 | 5 | 1.650 | 0 | 0.394 | 0.369 | 0.372 | 0.820 | -0.006 | 0.006 | 0.011 | 21.491 | 0.015 |
| MCW0081 | 0.494 | 42 | 11 | 3.001 | 1 | 0.667 | 0.560 | 0.622 | 1.483 | 0.126 | 0.156 | 0.034 | 7.140 | 0.000 |
| MCW0098 | 0.465 | 27 | 9 | 2.571 | 1 | 0.611 | 0.523 | 0.535 | 1.176 | 0.105 | 0.170 | 0.072 | 3.212 | 0.000 |
| MCW0103 | 0.708 | 9 | 6 | 1.736 | 2 | 0.424 | 0.375 | 0.349 | 0.693 | 0.131 | 0.160 | 0.033 | 7.343 | 0.000 |
| MCW0104 | 0.489 | 43 | 18 | 3.271 | 4 | 0.694 | 0.649 | 0.662 | 1.701 | 0.066 | 0.096 | 0.033 | 7.385 | 0.000 |
| MCW0111 | 0.595 | 21 | 8 | 2.440 | 0 | 0.590 | 0.483 | 0.550 | 1.226 | 0.110 | 0.141 | 0.035 | 6.800 | 0.000 |
| MCW0123 | 0.523 | 38 | 14 | 3.103 | 3 | 0.678 | 0.640 | 0.650 | 1.568 | 0.015 | 0.031 | 0.016 | 15.002 | 0.000 |
| MCW0165 | 0.635 | 7 | 4 | 1.924 | 0 | 0.480 | 0.302 | 0.386 | 0.755 | 0.325 | 0.341 | 0.024 | 10.050 | 0.000 |
| MCW0183 | 0.292 | 34 | 11 | 5.516 | 3 | 0.819 | 0.659 | 0.796 | 1.873 | 0.119 | 0.189 | 0.080 | 2.885 | 0.000 |
| MCW0206 | 0.394 | 24 | 9 | 3.992 | 2 | 0.750 | 0.699 | 0.714 | 1.583 | -0.004 | 0.044 | 0.048 | 5.000 | 0.000 |
| MCW0222 | 0.400 | 11 | 6 | 2.972 | 2 | 0.664 | 0.646 | 0.600 | 1.210 | -0.030 | 0.023 | 0.051 | 4.641 | 0.000 |
| MCW0248 | 0.679 | 6 | 4 | 1.816 | 1 | 0.449 | 0.492 | 0.366 | 0.713 | -0.236 | -0.185 | 0.041 | 5.864 | 0.344 |
| MCW0284 | 0.368 | 29 | 8 | 3.900 | 0 | 0.744 | 0.689 | 0.706 | 1.620 | 0.050 | 0.117 | 0.070 | 3.321 | 0.000 |
| MCW0295 | 0.465 | 34 | 13 | 3.482 | 3 | 0.713 | 0.579 | 0.680 | 1.632 | 0.131 | 0.214 | 0.096 | 2.341 | 0.000 |
| MCW0330 | 0.302 | 26 | 11 | 5.376 | 5 | 0.814 | 0.615 | 0.790 | 1.827 | 0.147 | 0.281 | 0.157 | 1.339 | 0.000 |
| Mean | 0.437 | 30.571 | 10.893 | 3.819 | 2.140 | 0.688 | 0.616 | 0.645 | 1.510 | 0.041 | 0.089 | 0.054 | 6.060 | |
| Total | | | 305 | | 60 | | | | | | | | | |

MAF, major allele frequency; NG, number of genotypes; NA, number of alleles; NPA, number of private allele; Ne, number of effective alleles; I, Shannon's information index; He, expected heterozygosity; Ho, observed heterozygosity; PIC, polymorphic information content, Nm: number of migrants, F, inbreeding coefficient over all populations ($F_{IS}$), among populations ($F_{IT}$) and within populations ($F_{ST}$), HWE pV, Hardy-Weinberg equilibrium p-value based on chi square test (There is a deviation from HWE at $p < 0.05$)

individuals from the SW population clustered together with the exotic chicken in group III, and finally the rest of SW individuals clustered with those from the CS in group II (Fig 3).

## Population structure

Data from the Bayesian cluster analysis showed the existence of four (4) main gene pools in the whole IC population in Rwanda. The highest value for *ΔK* was obtained for K = 4 (Table 8 and Fig 4). The first gene pool (I) was composed of CN and NW populations. The second gene pool (II) was made of the Eastern population only. The third (III) included individual from SW and CS and the fourth gene pool (IV) was composed of the remaining individuals of SW and exotic chicken. A high proportion of the admixture was observed in the gene pool III.

**Table 3. Common genetic diversity indices as revealed among IC populations in Rwanda.**

| Populations | N | %PL | NA | PA | Ne | Ho | He | uHe | F | I |
|---|---|---|---|---|---|---|---|---|---|---|
| Central North | 51 | 100 | 6.929 | 6 | 3.354 | 0.623 | 0.644 | 0.650 | 0.021 | 1.322 |
| Central South | 55 | 100 | 7.286 | 15 | 3.359 | 0.598 | 0.661 | 0.668 | 0.077 | 1.372 |
| Exotic chicken | 12 | 100 | 5.143 | 4 | 3.386 | 0.667 | 0.665 | 0.669 | -0.019 | 1.305 |
| East | 102 | 100 | 8.250 | 21 | 3.367 | 0.611 | 0.654 | 0.657 | 0.056 | 1.358 |
| North West | 52 | 100 | 6.500 | 0 | 3.311 | 0.613 | 0.645 | 0.651 | 0.042 | 1.306 |
| South West | 53 | 100 | 7.964 | 14 | 3.620 | 0.626 | 0.680 | 0.686 | 0.063 | 1.458 |
| Total | 325 | 100 | 7.011 | 60 | 3.400 | 0.623 | 0.658 | 0.668 | 0.040 | 1.353 |

N, Number of chickens, % PL, Proportion of polymorphic loci, NA, number of alleles; PA, number of private allele; Ne, number of effective alleles He, expected heterozygosity Ho, observed heterozygosity uHe: unbiased expected heterozygosity F, inbreeding coefficient I, Shannon's information index.

The results of the Factorial Correspondence Analysis (FCA) are depicted in Fig 5. It showed tree clusters whereby the Eastern region was still standing alone. NW and CN populations clustered together. Finally, the majority of individuals from the CS, SW and exotic chicken were in the same group.

**Table 4. Tests for the Hardy-Weinberg equilibrium probability of loci in the IC population in Rwanda.**

| Locus | North West | Central North | Central South | East | North-south | Exotic chicken |
|---|---|---|---|---|---|---|
| ADL0112 | 0.551 | 0.000 | 0.000 | 0.003 | 0.000 | 0.028 |
| ADL0268 | 0.000 | 0.000 | 0.163 | 0.000 | 0.000 | 0.330 |
| ADL0278 | 0.000 | 0.000 | 0.000 | 0.000 | 0.003 | 0.349 |
| LEI0094 | 0.001 | 0.976 | 0.000 | 0.051 | 0.001 | 0.812 |
| LEI0192 | 0.000 | 0.000 | 0.000 | 0.002 | 0.024 | 0.913 |
| LEI0234 | 0.000 | 0.000 | 0.000 | 0.099 | 0.000 | 0.720 |
| MCW0014 | 0.000 | 0.000 | 0.000 | 0.000 | 0.000 | 0.634 |
| MCW0016 | 0.012 | 0.000 | 0.000 | 0.239 | 0.108 | 0.200 |
| MCW0020 | 0.048 | 0.586 | 0.190 | 0.620 | 0.000 | 0.980 |
| MCW0034 | 0.050 | 0.735 | 0.316 | 0.000 | 0.816 | 0.412 |
| MCW0037 | 0.000 | 0.000 | 0.000 | 0.000 | 0.000 | 0.001 |
| MCW0067 | 0.000 | 0.000 | 0.870 | 0.000 | 0.000 | 0.095 |
| MCW0069 | 0.965 | 0.529 | 0.971 | 0.967 | 0.295 | 0.279 |
| MCW0078 | 0.911 | 0.251 | 0.985 | 0.232 | 0.003 | 0.916 |
| MCW0081 | 0.739 | 0.000 | 0.000 | 0.000 | 0.000 | 0.004 |
| MCW0098 | 0.681 | 0.000 | 0.000 | 0.000 | 0.000 | 0.005 |
| MCW0103 | 0.012 | 0.752 | 0.000 | 0.913 | 0.000 | 0.574 |
| MCW0104 | 0.001 | 1.000 | 0.355 | 0.000 | 0.755 | 0.213 |
| MCW0111 | 0.046 | 0.189 | 0.127 | 0.003 | 0.687 | 0.545 |
| MCW0123 | 0.503 | 0.909 | 0.000 | 0.002 | 0.000 | 0.003 |
| MCW0165 | 0.540 | 0.000 | 0.004 | 0.000 | 0.018 | 0.327 |
| MCW0183 | 0.000 | 0.010 | 0.000 | 0.000 | 0.012 | 0.001 |
| MCW0206 | 0.590 | 0.020 | 0.009 | 0.908 | 0.000 | 0.658 |
| MCW0222 | 0.000 | 0.096 | 0.000 | 0.783 | 0.968 | 0.283 |
| MCW0248 | 0.429 | 0.922 | 0.057 | 0.991 | 0.247 | 0.035 |
| MCW0284 | 0.121 | 0.021 | 0.846 | 0.000 | 0.000 | 0.437 |
| MCW0295 | 0.279 | 0.000 | 0.017 | 0.000 | 0.046 | 0.015 |
| MCW0330 | 0.633 | 0.992 | 0.000 | 0.000 | 0.150 | 0.001 |

P-values <0.05 show the genotype frequencies for nonconformity with Hardy-Weinberg Equilibrium (HWE) based on Chi square test

**Table 5. Analysis of molecular variance of all loci for the IC population in Rwanda.**

| Source | Degree of freedom | Sum square | Mean square | Estimated variances | % of estimated variances |
|---|---|---|---|---|---|
| **Among Populations** | 5 | 574.201 | 114.840 | 1.838 | 8% |
| **Within Populations** | 319 | 6346.643 | 19.895 | 19.895 | 92% |
| **Total** | 324 | 6920.843 | | 21.733 | 100% |

## Discussion

### Genetic diversity

The average PIC was the best index to estimate the polymorphism of alleles [16]. It showed that more information could be obtained from the loci when PIC>0.5. On the other hand, 0.25<PIC<0.5 was an indication of a moderately instructive locus, whereas PIC<0.25 indicated a vaguely informative locus [33]. In this study, 82.3% of all loci were highly informative, which confirmed that they were suitable for estimating the genetic diversity of IC populations in Rwanda. The highest value of PIC (0.87) was that of LEI0234 and the mean PIC was 0.6451. The PIC values found in this study exceeded those (0.29–080) of Cameroon's IC [17], and (0.31–0.49) of Chinese IC [7,8], but lower than those obtained by Tang for black-bone IC breeds (0.67) [34]. The mean frequency of alleles per marker found in this study (10.89) exceeded those recorded in previous reports in Cameroon (9.04) [17], Ghana (7.8) [35], Iran (5.4) [36], China (3.8) [37], Egypt (7.3) [38], Pakistan (9.1) [39] and Vietnam (6.41) [40]. The values obtained in this study were, however, lower than those from Brazilian (13.3) [41] and were in the same range as those from Ethiopian chicken ecotypes (10.6) [42].

The mean number of effective alleles (3.81) obtained was higher than 3.13 observed in Cameroon [17] and Indian chicken [21]. Heterozygosity can also be considered in genetic diversity. The degree of mean population heterozygosity is an indication of the level of population constancy. Low population heterozygosity informs high population genetic constancy [43]. The present study indicated that Ho of the different IC population varied from 0.3015 to 1 with an overall mean value of 0.6155, while He ranged from 0.394 to 0.887 with an overall average of 0.688.

This study also discovered that the values of Ho and He were similar. As a result, there was no significant difference between zero and the resultant F estimates (0.040), which suggested that the IC populations were in HWE. An implication of this supposition is that the population is under artificial selection, which is indicative of population stability. However, the little variation observed between Ho and He could be attributed to discrepancies in sample size, location, population composition, and the origin of microsatellite markers [44].

**Table 6. Genetic distance among the IC population in Rwanda.**

| Populations | North West | Central North | Central South | Exotic chicken | East |
|---|---|---|---|---|---|
| Central North | 0.029 | | | | |
| Central South | 0.094 | 0.077 | | | |
| Exotic chicken | 0.199 | 0.213 | 0.231 | | |
| East | 0.112 | 0.097 | 0.117 | 0.196 | |
| South West | 0.104 | 0.092 | 0.048 | 0.118 | 0.125 |

The extent of genetic distinction among the population with regard to allele frequencies ($F_{ST}$) and gene flow (Nm) are presented in Table 7. The results revealed a low genetic differentiation and a high gene flow between CN and NW, and likewise between SW and CS. A relatively high gene differentiation, however, was found between the E population and other populations.

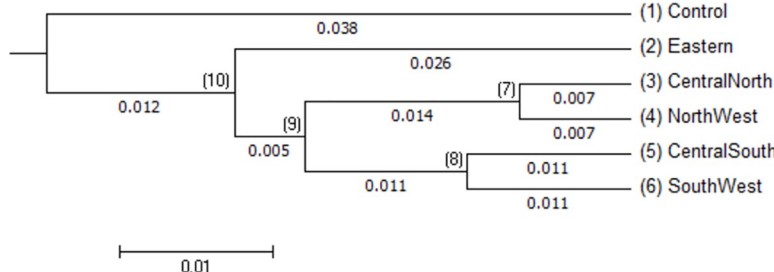

**Fig 2. Neighbour-Joining pair-wise of the IC population in Rwanda.**

The IC populations in Rwanda had a similar level of diversity as their Ethiopian [45], Egyptian [38] and Cameroonian [8] counterparts, but had lower and higher diversity than those observed in southern China [19], European and Asian IC breeds [35], respectively. Among Rwanda IC, all populations showed a significantly high degree of inbreeding, which could have an impact on trait fixation in the populations. This degree of inbreeding exceeded that observed for Yunnan IC breeds (0.25) [8] and Turkish IC (0.301) depicted with 10 SSR loci [44]. The $F_{ST}$ value (0.054) revealing the diversity between IC populations in Rwanda was higher than 0.048 for Ethiopian IC ecotypes [46] and (0.003–0.040) for Kenyan IC [47] and lower than 0.080 found in Cameroonian IC [17].

## Genetic relationships

Wright's F-statistics showing the inbreeding coefficient in this study was 0.041, which was higher than 0.03 found in Cameroon [17], but was similar to values obtained in many Chinese IC [18]. The $F_{ST}$ permits the approximation of migratory entities in a population per generation (Nm) based on loci. In IC populations in Rwanda, Nm varied from 1.332 to 21.491, with an average of 6.060. This value was higher than that obtained in Cameroun [17].

The number of private alleles (PA) distributed all through the ecotypes showed that there was high genetic diversity between populations. In this study, the number of PA was higher in the East (21) followed by CS (15) and SW (14). The NW population, however, did not exhibit

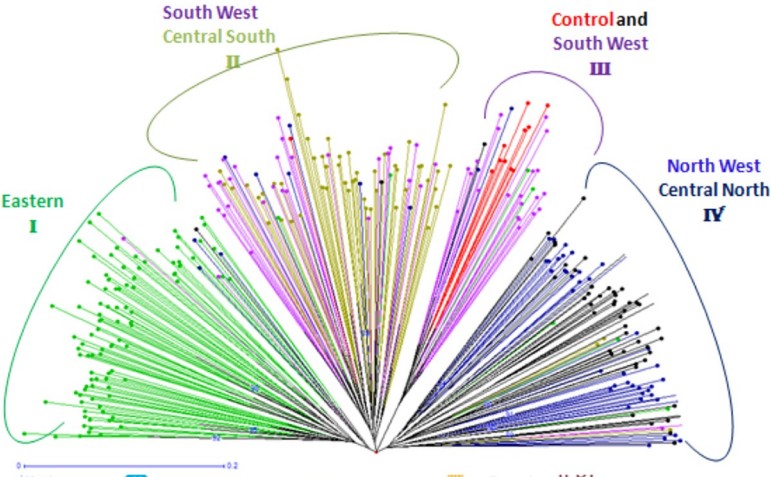

**Fig 3. Neighbour-Joining tree of the clustering pattern among IC populations in Rwanda.**

**Table 7. Gene flow (upper diagonal) and Gene differentiation (lower diagonal).**

| Populations | Central North | Central South | Exotic chicken | East | North West | South West |
|---|---|---|---|---|---|---|
| Central North | | 2.304 | 1.412 | 2.051 | 6.274 | 2.040 |
| Central South | 0.022 | | 0.925 | 1.471 | 1.533 | 3.847 |
| Exotic chicken | 0.052 | 0.058 | | 3.432 | 1.188 | 2.791 |
| East | 0.025 | 0.027 | 0.050 | | 1.783 | 1.560 |
| North West | 0.012 | 0.026 | 0.053 | 0.028 | | 1.471 |
| South West | 0.026 | 0.014 | 0.036 | 0.028 | 0.027 | |

any private allele (0). Despite, the number of private alleles being a good indicator of population relationship and structure, further studies need to be carried out to identify possible traits that may be controlled by these private alleles. The total number of private alleles in this study (60) was higher than that (24) found in Cameroun [17].

Findings from AMOVA showed that the largest portion of the genetic variation in IC populations in Rwanda existed in individuals within the population (92%). A comparable trend was noted in the Tanzanian [48], Ethiopian [17] and Cameroonian [17] IC ecotypes. The quality of the product, cultural uses of chicken, and the ease with which chicken adapts to the environment are the factors that motivate small-scale farmers to rear IC. These factors highlight the importance of within-population diversity as a key incentive in rearing IC [49].

Genetic distance within a population is a useful indicator of separation between various sub-populations. The key assumption of Nei's standard genetic distance is that hereditary dissimilarities are caused by mutations and genetic drift, whereas Reynolds distance assumes that the increase of genetic differences is due to genetic drift only [11]. The genetic distance between IC populations in SW and CS as well as between NW and CN were not significantly different (P>0.05). It was noted that these regions border each other, thereby implying that there is a high likelihood of sharing genetic materials. Another possible explanation is that these regions could be highly favorable to the IC population or IC populations in these regions

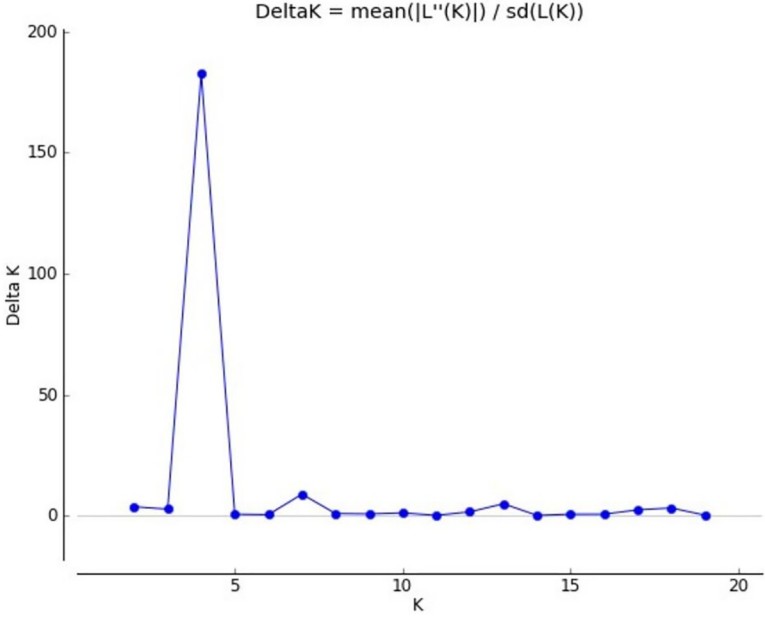

**Fig 4. Delta K (ΔK) approximating the more possible number of clusters in IC populations in Rwanda.**

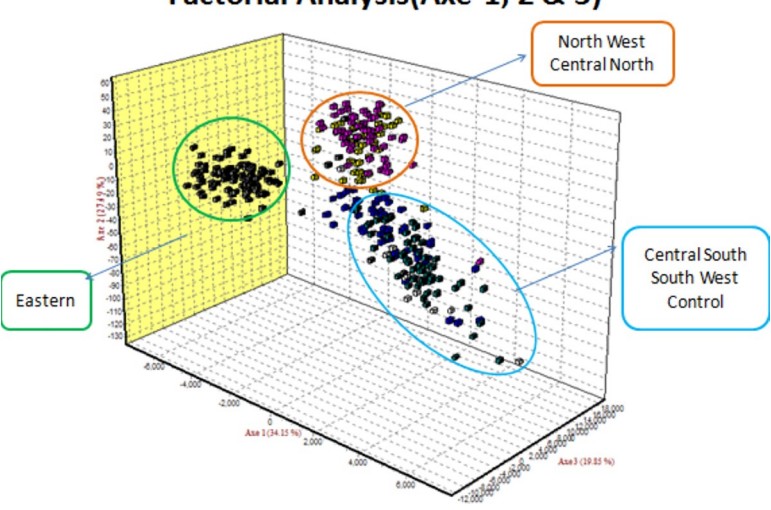

**Fig 5. Factorial correspondence analysis.**

could be big enough to prevent mutation and genetic drift. The genetic distances reported in this study fluctuated from 0.029 to 0.213. These values are in the range of those found in Egyptian IC [38] and in Chinese IC populations [50]. They are, however, higher than those observed in Chinese Bian chicken [19].

When estimating genetic differentiation using allele frequency in such scenarios, the genetic variance between populations can be explained by four major forces, namely, selection,

**Table 8. Number of clusters (K) based on the progression of the average estimate of Ln likelihood of data in IC populations in Rwanda.**

| K | Replication | Mean LnP(K) | Stdev LnP(K) | Ln'(K) | ILn"(K)I | Delta K |
|---|---|---|---|---|---|---|
| 1 | 5 | -27680.120000 | 0.192354 | - | - | - |
| 2 | 5 | -26645.700000 | 81.765916 | 1034.420000 | 301.520000 | 3.687600 |
| 3 | 5 | -25912.800000 | 30.968694 | 732.900000 | 82.920000 | 2.677543 |
| **4** | **5** | **-25262.820000** | **3.056469** | **649.980000** | **558.300000** | **182.661785** |
| 5 | 5 | -25171.140000 | 37.017671 | 91.680000 | 21.920000 | 0.592150 |
| 6 | 5 | -25057.540000 | 46.761341 | 113.600000 | 19.200000 | 0.410596 |
| 7 | 5 | -24963.140000 | 9.161496 | 94.400000 | 81.200000 | 8.863182 |
| 8 | 5 | -24949.940000 | 63.605566 | 13.200000 | 55.340000 | 0.870050 |
| 9 | 5 | -24881.400000 | 42.680968 | 68.540000 | 29.880000 | 0.700078 |
| 10 | 5 | -24842.740000 | 77.738491 | 38.660000 | 87.640000 | 1.127369 |
| 11 | 5 | -24891.720000 | 114.353824 | -48.980000 | 14.060000 | 0.122952 |
| 12 | 5 | -24954.760000 | 210.975195 | -63.040000 | 330.240000 | 1.565302 |
| 13 | 5 | -24687.560000 | 104.370245 | 267.200000 | 510.500000 | 4.891241 |
| 14 | 5 | -24930.860000 | 402.389690 | -243.300000 | 41.440000 | 0.102985 |
| 15 | 5 | -25132.720000 | 914.525050 | -201.860000 | 542.960000 | 0.593707 |
| 16 | 5 | -24791.620000 | 296.572178 | 341.100000 | 183.320000 | 0.618129 |
| 17 | 5 | -24633.840000 | 54.568333 | 157.780000 | 129.560000 | 2.374271 |
| 18 | 5 | -24605.620000 | 64.775126 | 28.220000 | 204.760000 | 3.161090 |
| 19 | 5 | -24782.160000 | 498.369745 | -176.540000 | 100.700000 | 0.202059 |
| 20 | 5 | -24858.000000 | 559.214181 | -75.840000 | - | - |

mutation, migration, and genetic drift [44]. Even though mutation plays a critical role in the long term, short-term evolution is mainly influenced by genetic drift in cases where populations segregated by reproduction [51]. Genetic distance analysis is used to show how close two populations are in relation to each other. The smaller the distance, the closer the two populations are to one another and vice versa [11]. IC populations showed segregation by distance and appeared to be at equipoise under the influence of dispersal and genetic drift. There is a high likelihood that these chickens were present at their current locations earlier than it had been assumed because there was not enough time for segregation due to distance to have come into play. Furthermore, long-distance gene dispersion is not satisfactorily evident to deter genetic deviation. For this, further investigations need to be conducted using more markers, for example, high-density SNP arrays and mitochondrial DNA which was also conducted concurrently with the current study.

## Phylogenetic relationship and population structure

The genetic similarity in a collection of breeds with high diversity can be resolved efficiently by cluster analysis, which facilitates the identification of individuals with similar or diverse multilocus genotypes [52]. A number of IC populations clustered together indicates genetic affinities between them [53]. In our study, the cluster based on the neighbour-joining approach revealed grouping arrays of association and genetic relationships among individuals. These individuals were grouped into four clusters formed by ecotypes from distinct collection sites (NW and CN; SW1 and CS; SW2 with exotic chicken and East alone). This close genetic relationship may indicate a common genetic background [54]. A cluster shows the degree of inbreeding and populations that could be sharing the identical ancestral lineage [55]. There is also similarities in morphological characteristics between the IC populations clustered together [56]. This was confirmed by the structure analysis which revealed four gene pools across IC in Rwanda. These gene pools are distributed exactly according to the different clusters as shown by the neighbour-joining method. The observed gene pools could be accounted for by the sum of private alleles recorded in the population besides the genetic distance between populations. For example, the Eastern region recorded the highest frequency of private alleles, whereas the NW had the lowest number. This observation could be attributed to the large population size of IC in the Eastern region out of all the study sites, which minimized gene inflow in this area. Conversely, the lowest number of IC was noted in the NW region, which could be interpreted to mean that the majority of chicken keepers in this area either buy chicken or exchange cocks from the neighbouring areas such as CN. Consequently, there is a high influx of genes in these regions. This is not surprising since these areas border each other geographically. These findings corroborated the observations of a study conducted in Kenya where the Mantel test had uncovered a positive association between hereditary and geographic distances [57]. Our study also confirmed that geographic distances affected the population's genetic structure [57]. The portion of SW chicken populations that clustered with the exotic chicken could be attributed to the fact that different crossing programmes between IC and improved chicken breeds have been introduced in that region to improve the genetic potential of IC in Rwanda [58].

## Conclusion

The results from this study are the first to recount the genetic diversity and constitution of IC from Rwanda. Overall, the IC populations in Rwanda had high levels of significant genetic variability as per different genetic diversity parameters applied in this study. Therefore, data on genetic diversity estimated by assimilating within and between population variances may inform preservation strategies and the better establishment of priorities. In addition, this study

found that IC in Rwanda belongs to four major gene pools that could be preserved independently to uphold their genetic diversity. Generally, these findings provide the fundamental step in the direction of judicious decision-making before the development of genetic enhancement and preservation programmes without interfering with the uniqueness of IC in Rwanda.

## Supporting information

**S1 Table. Characteristics of indigenous and exotic chickens used in the study.**
(DOCX)

**S1 Fig. Agro ecological zones in Rwanda.**
(PDF)

## Acknowledgments

The authors thank University of Rwanda, Rwanda Agriculture and Animal Resources Development Board and Egerton University for their technical support. We acknowledge also chicken owners for allowing collection of blood samples from their chicken.

## Author Contributions

**Conceptualization:** Richard Habimana, Tobias Otieno Okeno, Kiplangat Ngeno, Pauline Assami, Christian Tiambo Keambou, Nasser Yao.

**Data curation:** Richard Habimana.

**Formal analysis:** Richard Habimana, Pauline Assami, Anique Ahou Gbotto, Nasser Yao.

**Funding acquisition:** Richard Habimana, Pauline Assami, Nasser Yao.

**Investigation:** Richard Habimana, Christian Tiambo Keambou, Kizito Nishimwe, Janvier Mahoro.

**Methodology:** Richard Habimana, Tobias Otieno Okeno, Kizito Nishimwe, Nasser Yao.

**Project administration:** Richard Habimana, Pauline Assami, Nasser Yao.

**Resources:** Richard Habimana, Tobias Otieno Okeno, Kiplangat Ngeno, Pauline Assami, Christian Tiambo Keambou, Nasser Yao.

**Software:** Richard Habimana, Sylvere Mboumba, Anique Ahou Gbotto, Nasser Yao.

**Supervision:** Tobias Otieno Okeno, Kiplangat Ngeno, Nasser Yao.

**Validation:** Nasser Yao.

**Visualization:** Richard Habimana, Sylvere Mboumba, Christian Tiambo Keambou.

**Writing – original draft:** Richard Habimana.

**Writing – review & editing:** Richard Habimana, Tobias Otieno Okeno, Kiplangat Ngeno, Nasser Yao.

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
