## [Decision Letter · Decision Letter 0]

7 Jan 2020

PONE-D-19-29235

Genetic diversity and population structure of indigenous chicken in Rwanda using microsatellite markers

PLOS ONE

Dear Dr. HABIMANA,

Thank you for submitting your manuscript to PLOS ONE. After careful consideration, we feel that it has merit but does not fully meet PLOS ONE’s publication criteria as it currently stands. Therefore, we invite you to submit a revised version of the manuscript that addresses the points raised during the review process.

We would appreciate receiving your revised manuscript by Feb 21 2020 11:59PM. To enhance the reproducibility of your results, we recommend that if applicable you deposit your laboratory protocols in protocols.io, where a protocol can be assigned its own identifier (DOI) such that it can be cited independently in the future. For instructions see: http://journals.plos.org/plosone/s/submission-guidelines#loc-laboratory-protocols

We look forward to receiving your revised manuscript.

Kind regards,

Tzen-Yuh Chiang

Academic Editor

PLOS ONE

Journal Requirements:

2.  Thank you for including your ethics statement:  "Ethical clearance committee of the College of Agriculture, Animal Sciences and Veterinary Medicine, University of Rwanda.

Ref: 031/19/DRI September 2, 2019".

Please amend your current ethics statement to confirm that your named ethics committee specifically approved this study.

For additional information about PLOS ONE submissions requirements for ethics oversight of animal work, please refer to http://journals.plos.org/plosone/s/submission-guidelines#loc-animal-research  

3.  In your Methods section, please provide additional details regarding the chicken used in your study and ensure you have described the source. For more information regarding PLOS' policy on materials sharing and reporting, see https://journals.plos.org/plosone/s/materials-and-software-sharing#loc-sharing-materials.

4. In your Methods section, please provide additional location information of the sampling locations, including geographic coordinates for the data set if available.

5. In your Methods section, please provide additional information regarding the permits you obtained for the work. Please ensure you have included the full name of the authority that approved the sampling locations access and, if no permits were required, a brief statement explaining why.

Reviewers' comments:

Reviewer's Responses to Questions

**Comments to the Author**

1. Is the manuscript technically sound, and do the data support the conclusions?

Reviewer #1: Yes

Reviewer #2: Partly

2. Has the statistical analysis been performed appropriately and rigorously? 

Reviewer #1: Yes

Reviewer #2: No

3. Have the authors made all data underlying the findings in their manuscript fully available?

Reviewer #1: Yes

Reviewer #2: Yes

4. Is the manuscript presented in an intelligible fashion and written in standard English?

Reviewer #1: No

Reviewer #2: No

5. Review Comments to the Author

Reviewer #1: The manuscript reports on a study of genetic diversity and population structure of indigenous chicken in Rwanda using microsatellite markers. The results revealed that the considerable genetic diversity in indigenous chicken in Rwanda, which represents a crucial genetic resource that can be conserved or optimized through genetic improvement. However, the manuscript requires a minor revision before it can be considered further. Overall, I think that the manuscript will need attention in the following areas:

1. The writing of the introduction is poor. The introduction section only introduces the importance of indigenous chicken, and I suggest adding some information about the recent research of microsatellite markers in chickens.

2. In the materials and methods, the twelve exotic chickens (layers and broilers) as a reference data should have a detailed information.

3. Statistical analysis methods are not clearly presented, for example, p-value is mentioned without description of what kind of test. In addition, the p-value should be written consistently in the all manuscript.

4. Line 31: “chicken” should be “chickens”.

5. Line 243: “3,31” should be “3.13”.

6. Line 243: “drift-” should be “drift”.

7. Please carefully check the format of the manuscript, and make sure it fits the journal style.

Reviewer #2: The manuscript entitled “Genetic diversity and population structure of indigenous chicken in Rwanda using microsatellite markers” aimed to evaluate the genetic diversity of IC in Rwanda using microsatellite markers. This is a relevant study since the knowledge of animal genetic resources has become an important issue in order to avoid the genetic erosion. Also, the local chickens are an alternative to sustainable development of livestock. The article is interesting to the journal subject area. However, some points need to be clarified to improve the manuscript understanding are listed below:

1) Introduction: The authors should reformulate the introduction in order to improve the understanding.

For example, I could not understand the sentence: “More than 40% of households keep poultry out of which approximately 80% consists of indigenous chicken (IC).” (lines 50-51).

2) M&M:

Please, specify with more details sample the collection method and the ethical permission for collection. The sentences describing the genetic groups should contain only the breed´s name, number of samples used and also, the number of regions/flocks that samples were collected. I suggest that description of phenotypic traits in a separate table, for better understanding.

3) Discussion:

In general, the discussion is basically descriptive, improving the information of the results previously showed and comparing the results of this and other papers already published. There is no discussion in terms of phylogenetic relationships or evolution, neither about the genetic distance within the breeds studied, which weakens the impact of the paper. The manuscript subject is interesting, but needs to be improved.

The English written should be improved.

6. PLOS authors have the option to publish the peer review history of their article (what does this mean?). If published, this will include your full peer review and any attached files.

Reviewer #1: No

Reviewer #2: No

---

## [Author Response · Author response to Decision Letter 0]

25 Feb 2020

Academic editor:

Ethic statement has been amended> Amended ethical clearance has been put in the methods section and also added to the Ethics Statement field of the submission form. Additional information of the sampling locations, including geographic coordinates for the data set were provided in S2 Fig and Fig 1. Additional information regarding the permits I obtained for the work has been provided and the full name of the authority that approved the sampling locations access was given

Reviewer #1:

Introduction has been improved and some information on microsatellite markers has been included in the introduction section (line 61-72). Detailed information on exotic chicken has been discussed and more details were put in supporting information (S1 Table). Statistical analysis has been improved. Concerning tests for the Hardy-Weinberg Equilibrium (HWE), P-value was obtained based on chi square (χ2) test and the nonconformity with HWE based on Chi square test at p<0.05. In addition, p-value has been written consistently throughout the manuscript. The introduction has been reformulated as per your request and it is now understandable. That confusing sentence has been paraphrased and it is now “ More than 40% of households keep poultry with indigenous chickens being the most preferred, accounting for approximately 80% of the reared chicken species”.The collection method has been detailed and ethical statement included.

Chickens, Corrected now line 26

3.13, Corrected now line 307

drift, Corrected now line 350

Reviewer #2:

The introduction has been reformulated as per your request and it is now understandable. That confusing sentence has been paraphrased and it is now “ More than 40% of households keep poultry with indigenous chickens being the most preferred, accounting for approximately 80% of the reared chicken species” . The collection method has been detailed and ethical statement included.

The genetic groups were named based on the agro-ecological zone of origin. In total, 313 IC were sampled from five agro-ecological zones (S2 Fig) and in each region, a specific number of IC was collected according to the size of the region. Phenotypic data was provided in supporting information (S1 Table). You are right. The discussion on phylogenetic relationship or evolution was under the subtitle called population structure; that is why it did not come out clearly. So as to make it clear, apart from its improvement, the subtitle has also been changed into “Phylogenetic relationship and Population structure”. 

The discussion on genetic distance is under “genetic relationship subtitle” and it has been improved as per your request.

The English written has been improved as you can see that thru the track changes.

---

## [Decision Letter · Decision Letter 1]

10 Mar 2020

Genetic diversity and population structure of indigenous chicken in Rwanda using microsatellite markers

PONE-D-19-29235R1

Dear Dr. HABIMANA,

We are pleased to inform you that your manuscript has been judged scientifically suitable for publication and will be formally accepted for publication once it complies with all outstanding technical requirements.

With kind regards,

Tzen-Yuh Chiang

Academic Editor

PLOS ONE

Additional Editor Comments (optional):

Reviewers' comments:

Reviewer's Responses to Questions

**Comments to the Author**

1. If the authors have adequately addressed your comments raised in a previous round of review and you feel that this manuscript is now acceptable for publication, you may indicate that here to bypass the “Comments to the Author” section, enter your conflict of interest statement in the “Confidential to Editor” section, and submit your "Accept" recommendation.

Reviewer #2: All comments have been addressed

2. Is the manuscript technically sound, and do the data support the conclusions?

Reviewer #2: Yes

3. Has the statistical analysis been performed appropriately and rigorously? 

Reviewer #2: Yes

4. Have the authors made all data underlying the findings in their manuscript fully available?

Reviewer #2: Yes

5. Is the manuscript presented in an intelligible fashion and written in standard English?

Reviewer #2: Yes

6. Review Comments to the Author

Reviewer #2: Most of the suggestions were addressed and now the paper is acceptable for publication. There are some spelling errors, which could be corrected in the proofs.

7. PLOS authors have the option to publish the peer review history of their article (what does this mean?). If published, this will include your full peer review and any attached files.

Reviewer #2: No

---

## [Editor Report · Acceptance letter]

12 Mar 2020

PONE-D-19-29235R1 

Genetic diversity and population structure of indigenous chicken in Rwanda using microsatellite markers 

Dear Dr. HABIMANA:

I am pleased to inform you that your manuscript has been deemed suitable for publication in PLOS ONE. Congratulations! Your manuscript is now with our production department. 

With kind regards,

on behalf of

Dr. Tzen-Yuh Chiang 

Academic Editor

PLOS ONE